# Autonomous Visual Fish Pen Inspections for Estimating the State of Biofouling Buildup Using ROV

Matej Fabijanić *, Nadir Kapetanović * and Nikola Mišković *

Faculty of Electrical Engineering and Computing, University of Zagreb, Unska 3, 10000 Zagreb, Croatia
* Correspondence: matej.fabijanic@fer.hr (M.F.); nadir.kapetanovic@fer.hr (N.K.); nikola.miskovic@fer.hr (N.M.)

**Abstract:** The process of fish cage inspections, which is a necessary maintenance task at any fish farm, be it small-scale or industrial, is a task that has the potential to be fully automated. Replacing trained divers who perform regular inspections with autonomous marine vehicles would lower the costs of manpower and remove the risks associated with humans performing underwater inspections. Achieving such a level of autonomy implies developing an image processing algorithm that is capable of estimating the state of biofouling buildup. The aim of this work is to propose a complete solution for automating the said inspection process; from developing an autonomous control algorithm for an ROV, to automatically segmenting images of fish cages, and accurately estimating the state of biofouling. The first part is achieved by modifying a commercially available ROV with an acoustic SBL positioning system and developing a closed-loop control system. The second part is realized by implementing a proposed biofouling estimation framework, which relies on AI to perform image segmentation, and by processing images using established computer vision methods to obtain a rough estimate of the distance of the ROV from the fish cage. This also involved developing a labeling tool in order to create a dataset of images for the neural network performing the semantic segmentation to be trained on. The experimental results show the viability of using an ROV fitted with an acoustic transponder for autonomous missions, and demonstrate the biofouling estimation framework's ability to provide accurate assessments, alongside satisfactory distance estimation capabilities. In conclusion, the achieved biofouling estimation accuracy showcases clear potential for use in the aquaculture industry.

**Keywords:** fish cage inspection; aquaculture biofouling estimation; underwater image segmentation; autonomous ROV control loop; image annotation tool

## 1. Introduction

There has been a growing acknowledgment of the crucial role played by small-scale fisheries and industrial aquaculture in ensuring global food security and nutrition in the 21st century. Global aquaculture production has shown a rising trend over the last 30 years, with around 80 million tonnes of seafood produced in 2020 [1]. Aquaculture is known to be highly labor-intensive, requiring significant human involvement in various tasks such as feeding, cleaning and processing as opposed to the envisioned streamlined machine-supported industrial agriculture. The emergence of autonomous robots presents an opportunity to supplement and enhance these labor-intensive operations. By integrating autonomous robots, the aquaculture sector can achieve higher efficiency, reduce operational costs, and ensure sustainable fishing practices for the future.

Previous research in this field often deals with only one specific aspect of aquaculture activities, like net damage detection in [2,3]. Other work such as that by Duda et al. [4] uses computer vision techniques to achieve ROV pose estimation and briefly touches on biological buildup on the net, but only mentions it as a potential problem for pose estimation. More work by Livanos et al. [5] discusses enhancing ROV autonomy level through intelligent navigation, but does not showcase a use for any specific fishery maintenance

task. Work by Qiu et al. [6] examines the estimation of built-up biofouling using image processing, but only briefly mentions using an ROV to capture footage needed for research. To the best of the authors' knowledge, there is no literature on the development of a complete autonomous fish cage inspection system that includes not only visual biofouling estimation, but also control and localization of an underwater vehicle. A research project named HEKTOR (Heterogeneous autonomous robotic system in viticulture and mariculture) aimed to fill this knowledge gap and offer a solution that enables efficient coordination among heterogeneous autonomous robots, as can be seen in Figure 1. More information about the project can be found in [7,8].

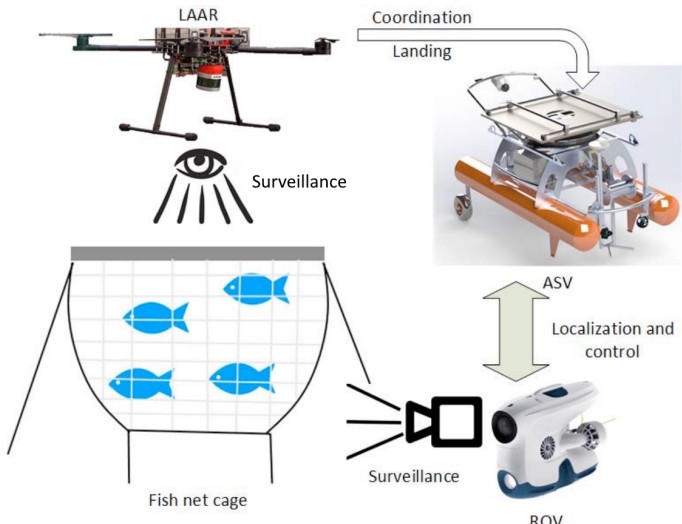

**Figure 1.** A schematic of the proposed HEKTOR underwater inspection solution.

One task that can, or better said should be automated, is the inspection of fish cage nets in aquaculture. The net gradually accumulates biofouling which has many negative consequences. The main problem is the reduction in available area for clean water to flow through which causes the water inside the pen to become less oxygenated and more fouled, ultimately resulting in increased fish sickness and death rate. Accompanying problems include the addition of extra mass to the pen structure, thus placing stress on the mooring ropes, damaging the net and causing a need for reknitting it. Periodic visual inspections by divers are currently necessary to assess the condition of the pens and determine the appropriate timing for cleaning. Using an ROV or an AUV to perform said visual inspections would reduce the need for divers, thus creating a more streamlined, autonomous, and risk-averse inspection process. It is worth mentioning publications from the scope of the HEKTOR project related to this topic. One of them touched on the topic of pose estimation published in [9] that compares two different visual servoing approaches. Another paper diving into more detail about semantic image segmentation and distance estimation can be found in [10], which directly precedes the work described in this paper.

As shown in Figure 1, in the scope of the HEKTOR project, a completely automated underwater fish cage inspection process includes an autonomous surface vehicle (ASV) integrated with an ROV. This integration means that the camera is streamed from the ROV, while biofouling estimation, control, and underwater algorithms are run onboard the ASV. The inspection process is envisioned to be split into two tasks. The first task is processing the underwater footage obtained from a filming vehicle in such a way to correctly estimate the amount of biofouling present on the net. The second task is developing an autonomous control algorithm to maneuver an ROV around the pens that is controlled via an ASV.

The main research goal was to develop an image processing algorithm that could be combined with the control algorithm in order to accurately assess the amount of biofouling accumulated on the nets. This research paper contributes in several key areas. Firstly, a neural network is successfully utilized for accurately segmenting underwater images of

fish pens, enabling precise identification of pen structures. The architecture used for AI segmentation is the popularly employed UNet [11], explained in more detail later. Secondly, the research explores and tests the feasibility of retrofitting an ROV with an underwater transponder to achieve precise localization. This modification improves upon its manual operation while also enabling autonomous missions, enhancing both its operational ease and versatility. Thirdly, the paper proposes a technique for estimating the extent of bio-fouling buildup on a given pen. By combining image segmentation with localization, the technique provides a valuable means of quantifying the level of biofouling on fish pens. Lastly, the proposed biofouling estimation technique is successfully tested in controlled and repeatable experimental scenarios. Collectively, these contributions improve fish pen bio-fouling analysis, and highlight potential advancements in the field of underwater robotics and mariculture practices.

The rest of this article is organized as follows: Section 2 discusses both the acquired and developed equipment used during the research, with two subsections that examine the custom-made ASV, and retrofitting of the commercially available ROV in detail. Section 3 presents the process of collecting underwater footage of fish pens, describing the methods employed to capture the necessary footage used in the research. The developed labeling tool is then described and the annotation process is presented. In addition, the approach used for estimating the amount of biofouling is outlined. Furthermore, the developed ROV control algorithm is presented, and its development and implementation is elaborated upon. Section 4 describes the experimental setup used to validate the biofouling estimation algorithm performance. Results obtained from the conducted experiments are presented in Section 5. Concluding remarks and future work are outlined in Section 6.

## 2. Equipment

### 2.1. ASV Korkyra

A custom-made aluminum catamaran named Korkyra was developed to function as a versatile remote-controlled or autonomous surface vehicle, as shown in Figure 2. Detailed information concerning the development of the catamaran can be found in [7,12]. This specially designed catamaran boasts several key features, one of which is a landing platform dedicated to accommodating a lightweight drone for aerial operations [13]. Additionally, it incorporates a docking and tether management system intended for seamless integration with an ROV, enabling underwater mission capabilities, [14]. It also features a robust metal frame that enables the mounting of diverse tools and sensors such as cameras, sonar, lidar, etc.

To support real-time control algorithms, data processing, and other computational requirements, a powerful onboard computer is used, ensuring efficient and reliable operation. With its purpose-built design and advanced functionalities, the custom aluminum catamaran serves as a valuable asset for remote-controlled or autonomous operations, facilitating enhanced surveillance, data collection, and exploration capabilities both on and under the water surface.

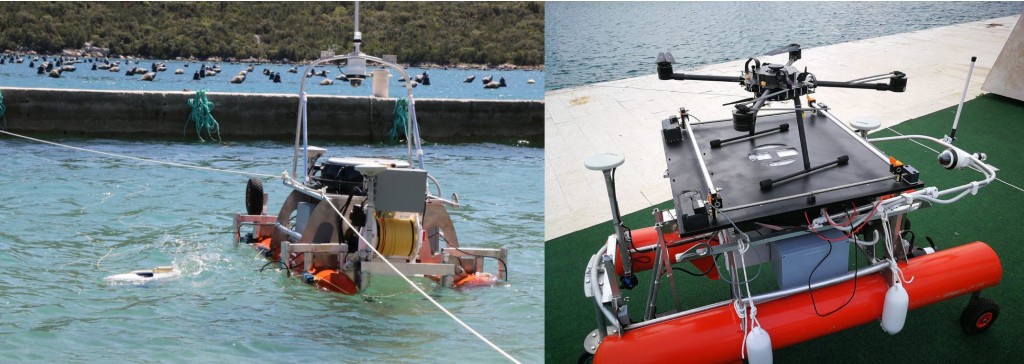

**Figure 2.** Autonomous surface vehicle Korkyra: (**left**)—TMS mounted, (**right**)—LP mounted onto the ASV.

### 2.2. ROV and Acoustic Localization System

Accurate localization in a cluttered environment such as fisheries is a critical component for achieving autonomy. In order for autonomous systems to navigate and operate effectively in such environments, they must be able to precisely estimate their own position relative to obstacles, structures, and other objects. Accurate localization enables the autonomous control system to make informed decisions, plan optimal paths, and avoid collisions. An underwater acoustic positioning system is essential for autonomous underwater missions due to the absence of GPS and standard RF-based positioning systems in underwater environments. This technology plays a vital role in enabling autonomous underwater missions by providing reliable positioning information where traditional positioning systems cannot operate effectively [15]. The use of acoustic signals allows for precise localization and tracking of AUVs and ROVs or other underwater assets. By using acoustic signals, the positioning system enables accurate navigation, mapping, and control of underwater vehicles in real-time.

A commercially available ROV was acquired from Blueye, a Norwegian company specializing in underwater technology. The ROV was mounted with a transponder belonging to an SBL acoustic underwater positioning system, as can be seen in Figure 3 below. The SBL acoustic localization system is the Underwater GPS G2 acquired from WaterLinked, also a Norwegian company specializing in acoustic subsea communication and positioning systems. An L-shaped fixed position configuration with 4 transceivers placed at various depths, and 1 transponder mounted on the ROV was precise enough for use in a controlled test inspection scenario. A secondary, but perhaps more realistic rectangular configuration with the 4 transceivers mounted on an ASV was also tested and provided good results in rough weather conditions. However, the depth reading was taken directly from the ROV because the sensor readings were far more precise than those received from the SBL transponder. More information about the integration process, underwater acoustical positioning system, accuracy testing and validation can be found in [16].

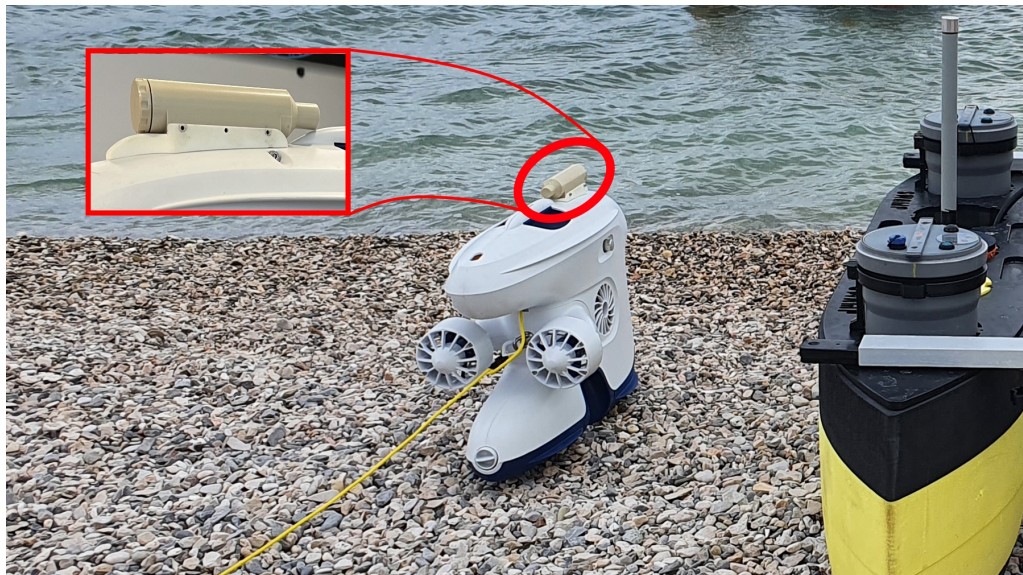

**Figure 3.** The used Blueye Pro ROV and a closeup of the retrofitted WaterLinked Underwater GPS G2 transponder.

## 3. Methods

### 3.1. Dataset Collection and Labeling

The crucial initial step of this research involved using a manually controlled ROV to collect underwater video footage of circular fish cages utilized in fish farming during the summers of 2020 and 2021. High-quality video footage of real industrial fish pens was filmed in the Adriatic Sea near the coast of the island of Ugljan. The collected underwater

footage showcased the net cages in various states of biofouling buildup, ranging from relatively clean to those in need of cleaning. This diverse range of footage provided valuable visual documentation of the gradual accumulation of biofouling on the fish cages over time. The footage gathered during those two years served for all computer vision-related research leading up to trials and validation experiments in a controlled seawater pool in Biograd, Croatia in late September and early October of 2022. There was close to 4000 images available from Ugljan for the creation of an initial dataset, and each mission in Biograd generated close to 1000 images. Later in this paper, a more detailed breakdown of how many images were labeled and ended up in the final dataset will be presented in Section 5.2.

The first step in the process of estimating the amount of biofouling built up on the pen net involves extracting useful information from underwater footage. This poses a well-known image segmentation problem. To solve it, it was first necessary to create a dataset of labeled images of the fish cages. To efficiently label hundreds or even thousands of images, a labeling tool was developed. This tool utilized the K-Means clustering algorithm [17] to group pixels of similar colors within the images. In short, K-Means clustering is an unsupervised machine learning algorithm used for grouping vectors of the same size into clusters based on their similarity, with the K being a hyperparameter representing how many clusters we expect to find. It works by generating K random starting centroids, then iteratively assigning vectors to their nearest centroid, and then updating the centroids to minimize the total sum of squared distances between vectors and their respective centroids.

By segmenting the images based on color similarities, the labeling tool significantly expedited the process of labeling the cage structure. This approach made it possible to handle large volumes of images effectively, streamlined the labeling process and enabled the creation of comprehensive datasets for further image analysis and training of neural networks used for semantic image segmentation. The tool, shown in Figure 4, works under the assumption that underwater images naturally lose yellow–orange hues as depth increases, with the dominant colors in images being blue, brown and green. This makes it possible to faithfully simplify an image down to less than 10 colors, while not losing much structural detail. An operator can select which image needs annotation, choose the color space (either LAB or HSV if LAB is not satisfactory), and choose the K parameter (mostly set to 7). This is all done through the toolbar shown at the top of Figure 4. The entire image is used as the input into the K-Means algorithm, and the result is that similarly colored pixels belong to the same cluster. The clustered pixels also most often belong to the same entity, that is, they have the same semantic meaning. The tool can turn the pixels in an image belonging to a cluster on or off so that the operator can easily see to what object on the image they belong to. The labels assigned to each pixel were either "sea", "cage", "fish", and "blurry", as shown in the bottom left of Figure 4.

### 3.2. Biofouling Estimation Framework

The modular biofouling estimation framework proposed herein is depicted in Figure 5. The framework comprises several interconnected nodes with each node serving a specific purpose. The framework and the nodes came about as a product of developing image processing algorithms and the ROV control loop in the Robotic Operating System (ROS). Node (1), shown in Figure 5, is responsible for executing the image segmentation process, separating the fish pen structure and its net from the background. Node (2) combines the segmented data with the distance information to reconstruct how a clean net would appear at that specific filming distance. Node (3) compares the two binary images, one of the ideal net state and one of the current state, and the result quantifies the extent of biofouling coverage on the net's surface. Node (4) implements pose (distance) estimation from a single camera if the distance from the filmed net is not known from a 3D map of the environment or some other source.

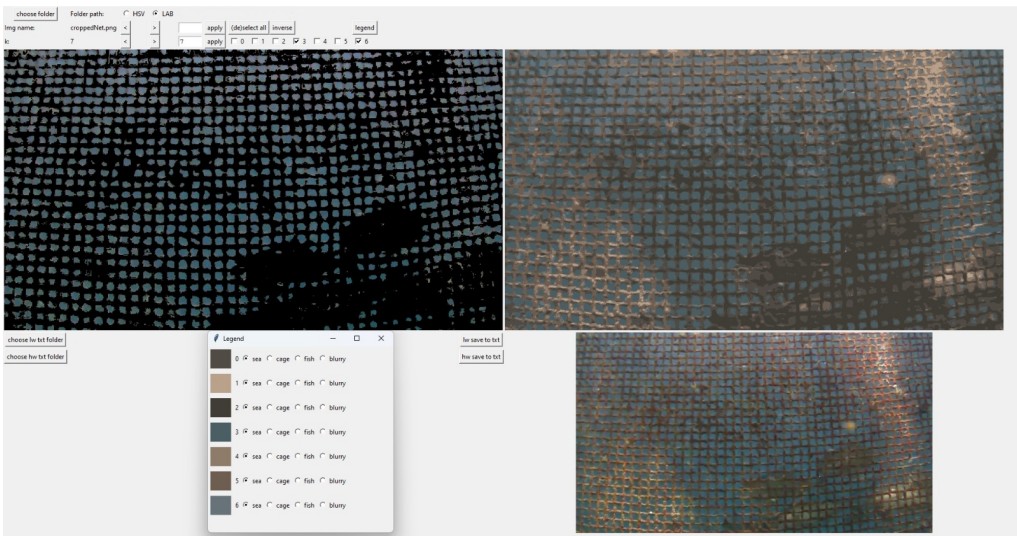

**Figure 4.** Screenshot of the labeling tool developed for easier dataset creating. The image in the top left is the original image with clusters of pixels turned on or off. The image in the top right replaces pixels from the original image with their respective centroids resulting from K-Means clustering. It is possible to choose the color space of the image, change the K hyperparameter, and turn the pixels on and off using the toolbar above the images. The legend pop-up window in the bottom left is used for turning the grouped pixels "on or off" and assigning labels. The color squares represent the resulting centroids of K-Means clustering.

A particular implementation of the framework developed during research calculates an estimated distance from the net by determining the approximate distance between each center of the small squares in the net, shown in more detail in Figure 6. The selection of centers as good features in the biofouling estimation process was based on their inherent stability. As biofouling builds up on the net, the squares gradually reduce in size, however the position of the center remains constant. This property makes the centers robust features for detection. By having knowledge of the distance in pixels for specific features on an object in an image, as well as the corresponding real-life distances, along with information about the camera sensor used for capturing the picture such as the physical size of the sensor and its focal length, it is possible to estimate the distance from the observed object to the camera. This allows for a reasonable estimation of the distance from the camera to the observed fish pen. This method of distance estimation can be classified as a geometric approach to the problem, as opposed to using a more time-consuming and computationally heavier deep-learning approach.

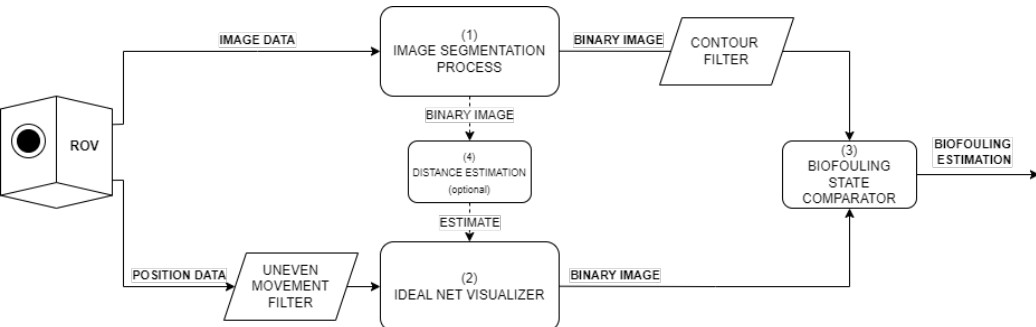

**Figure 5.** Schematic of the used biofouling estimation framework.

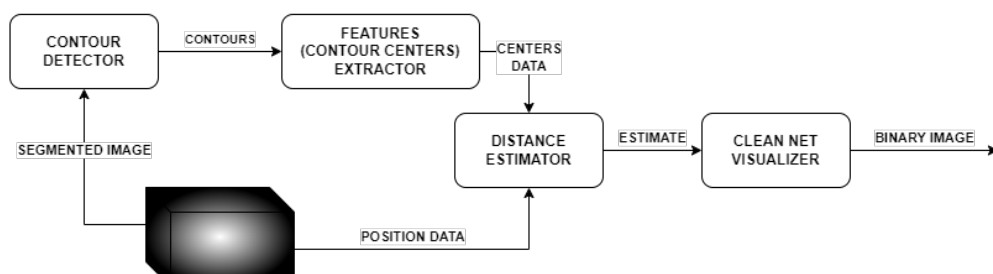

**Figure 6.** Ideal net visualizer part of the framework shown in more detail as it is implemented.

### 3.2.1. Biofouling Buildup Quantification

The described framework deals with one image at a time. Two different approaches were tested during research concerning how to estimate the total amount of biofouling for a filmed net. The idea behind an initial approach was to have an ROV film the fish pen net from very specific positions, chosen mathematically to have the least amount of overlap between any two images. These images could then be combined into a "mosaic" of the complete fish pen net and ran through the estimation framework. The problem with this approach is that it would require very precise localization data and an always up-to-date precise 3-D map of the environment so that the ROV can be localized with regard to the fish pens. Such precise localization and a 3-D map was not available during testing, which produced unsatisfactory results shown later.

A more realistic approach would be to film the net area of interest with as uniform a movement as possible at a fixed distance from the net. Uniform movement of the ROV would ensure that each segment of the net is filmed for roughly the same amount of time. If this is the case, then the images that go into the framework do not have to be picked because every area of the net is filmed for the same amount of time. The entire footage can be fed into the framework, and the amount of biofouling for the filmed area would be the average of all of the estimations. This approach was used for obtaining accurate results that are presented later, albeit the footage was decimated so that ROV would save its current high resolution frame every second due to storage restrictions. The ROVs movement speed was such that filming at 1Hz was frequent enough to capture the entire net area.

### 3.2.2. Framework Filters

There are two filters seen in Figure 5. The "uneven movement filter" node utilizes estimated distance data to exclude any footage captured during periods of unstable movement, ensuring that only reliable footage is considered. This approach works under the assumption that the ROV moves uniformly during autonomous missions. The filter tries to detect sudden changes in movement and invalidate any footage filmed during that specific time period. This filter can be applied both in real-time or offline after completing a mission. Figure 7 visualizes how the filter works, with footage that is captured during stable movement marked with green, and footage captured during non-uniform movement marked red.

The "contour filter" node (see Figure 5) uses a detected contour shape and area filter. In short, it is possible to detect square contours of the net openings in the segmented images using standard contour detection techniques. Furthermore, it is possible to filter out any contours deemed unwanted using their features such as shape or area. It is reasonable to assume that the detected contours should be of square shape, and roughly the same size. If some contour is too small or large, or not square in shape, then it can be discarded in further processing. More details on contour detection can be found in [10].

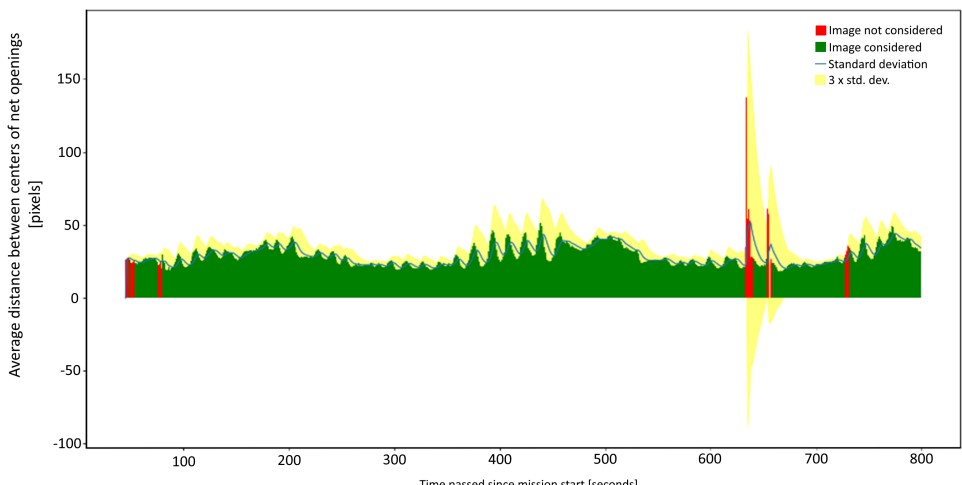

**Figure 7.** Plot visualizing the calculated average distance of centers of net openings for one mission. The green bar represents an image that passed through the movement filter, while the red bar represent images that are deemed to have been captured during periods of non–uniform movement.

### 3.2.3. Image Segmentation Node

The most computationally heavy node, node (1), shown in Figure 5, dealing with image segmentation underwent a major revision during research. At first, an approach involving machine learning was tried before using neural networks. The resulting data from labeling include the semantic meaning of not only pixels, but also colors within the image since it involved differentiating the boundary between cage structure, background sea, and fish using K-Means. This provided an opportunity for further analysis and exploration of color-related characteristics within the labeled images. The labeled colors obtained from the image labeling process were utilized to train a logistical regression model. This model used the numerical color representation to learn the relationships between different colors and their corresponding semantic labels, meaning whether a specific pixel belongs to the pen structure or the background sea and fish. The dataset was split into the standard 80–20% train–test split, meaning that the model weights were trained on data stemming from 4 out of 5 images, and the 5th image was used for testing. More on this method can be found in [10]. This initial approach was abandoned due to its lack of robustness and limited performance. While it provided fine results on images used to train it, it struggled to handle variations in lighting conditions, color temperature, inherent blurriness associated with images taken while moving, etc. As a result, a more advanced and robust approach was sought to overcome these limitations and improve the accuracy and reliability of the semantic segmentation node.

Utilizing neural networks for image segmentation is a more robust approach due to their ability to learn complex patterns and features. Networks like these can adapt to variations in lighting, color temperature, etc. This makes them well-suited for the task of underwater semantic segmentation. The neural network architecture chosen for this task is UNet, a well-established and widely used model in the field of image segmentation. Originally developed for semantic segmentation of medical images, UNet's complex architecture makes it a good choice for a wide range of segmentation tasks. UNet received its name from its characteristic U-shaped architecture. This neural network architecture functions by first encoding an input image through a series of convolutional and pooling layers to capture features and context. It then uses skip connections to combine features from the encoding path with those from the decoding path, which consists of upsampling and convolutional layers, to produce a pixel-wise segmentation mask, classifying each pixel into one of several predefined categories. This U-shaped architecture allows UNet to effectively capture both high-level context and fine-grained details for accurate image

segmentation [11,18]. The UNet model was trained to distinguish between the net of the fish pens and the background behind it showing sea and fish. The dataset was again split into the standard 80–20% train–test split, but this time the image sets were chosen randomly from the entire image pool.

The only image processing done before feeding the image to the AI model for segmentation is sharpening using the standard Laplacian sharpening filter. Other popular image processing techniques like blurring or histogram equalization were tried, but abandoned due to unsatisfactory results.

### 3.3. Closed-Loop ROV Control System

Initial work carried out involving autonomous control with the acquired ROV involved the task of maintaining a straight path while ascending and descending along a fish pen section using only visual odometry. This work aimed to test image processing techniques for the specific underwater inspection use-case and develop an algorithm that allowed the ROV to accurately perceive its position and orientation relative to the pen structure, enabling smooth and precise navigation in vertical movements [19]. Further advancements in achieving autonomous control using visual odometry relied on detecting distinctive feature points within the captured images. These feature points were then utilized in conjunction with established image processing techniques to estimate the movement of the ROV from a single camera [9]. Seeing how the ROV is now retrofitted to incorporate precise underwater localization, the next step was to use the available localization data in control algorithms.

A lawnmower pattern trajectory controller was designed to serve as a preliminary proof-of-concept test to validate the possibility of complete autonomous control for the modified ROV. This test aimed to assess and confirm crucial functionalities such as precise position estimation, efficient trajectory planning, and most importantly to validate the responsiveness of the ROV's control loop. Successful execution of the lawnmower pattern trajectory test would serve as a confirmation of the feasibility of autonomous control for the modified ROV and provide a solid foundation in further developing and refining the ROV's autonomous capabilities for more intricate tasks in real-world scenarios. It would pave the way for using autonomous control to enhance efficiency, safety, and productivity in underwater operations of fisheries, opening new possibilities for further research, and industrial applications.

The control algorithm for the ROV was implemented using multiple states to facilitate its movement [20]. Each state represents a specific behavior or action for the ROV, such as moving to a starting position, swaying, descending, or resurfacing to a predetermined end point. A general schematic for the control system can be seen in Figure 8, and the different states of the controller can be seen in a UML class diagram of the controller implementation in Figure 9 below. The control loop is closed by using the position data from the UWGPS transponder mounted directly onto the ROV to compare current position with the active waypoint. The control system implemented for the ROV managed surge and heave motions. The control algorithm should effectively control and stabilize these motions to ensure precise positioning and maneuverability underwater [21]. Additionally, the ROV features built-in automatic heading maintenance, that is, it can travel in a constant direction during operation. This feature was turned on during missions. The position error is used as an input into a classic PID controller that generates thruster commands, so that the control system can be used for any 2-DOF ROV [22]. The thruster command is calculated using the standard PID formula: $u(t) = K_p e(t) + K_i \int_0^t e(t) dt + K_d \frac{de}{dt}$, where $K_p, K_i, K_d$ are the tunable gain parameters, and $e(t)$ is the position error. Since underwater localization systems often have considerable signal noise, the gain for derivative term was set to $K_d = 0$ to avoid random thruster force spikes.

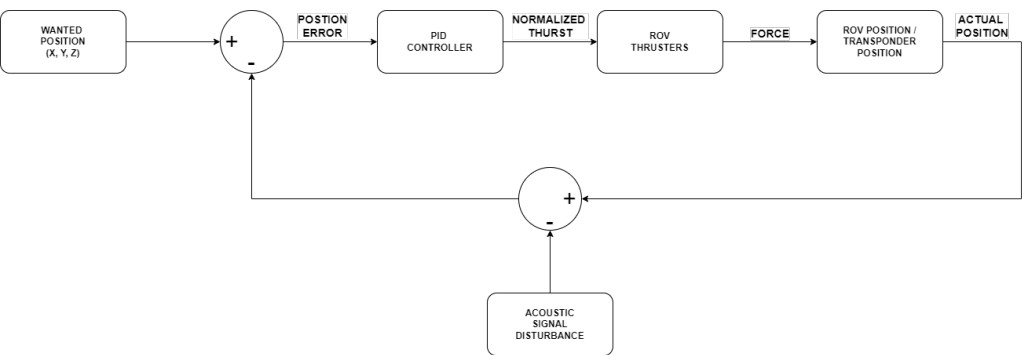

**Figure 8.** Schematic depicting the ROV control loop.

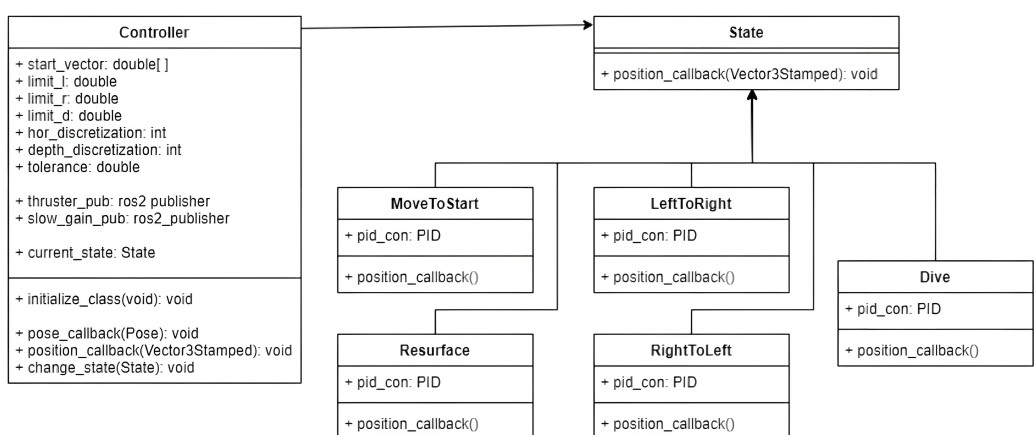

**Figure 9.** Schematic showing the implemented control loop class diagram. The "Controller" class generates waypoints by taking into account the leftmost and rightmost possible values for the position, depth, and how many discrete points to generate along the horizontal and vertical axes. The outputs are values for thruster speed along two controlled degrees of freedom (surge and sway).

## 4. Experimental Setup

Validation experiments for the biofuling estimation framework were conducted in a controlled seawater pool in Biograd, Croatia in late September and early October of 2022. For the purpose of conducting these trials, a pen net was acquired from an industrial fish farm and deployed within the controlled environment of an Olympic-sized pool 50 m long and 25 m wide. Only half of the total pool length was used for testing, so the actual experiment work area was around 625 m$^2$.

The dimension of the net is ∼14 m wide and ∼3 m high, so ∼42 m$^2$ of area in total. This acquisition enabled the emulation of real-world conditions and realistic performance evaluation of the developed system. To simulate biofouling buildup in this scenario, camouflage-pattern colored square patches were strategically hand-placed onto the fish cage net as can be seen in Figure 10. The patches were 0.25 m squares. They were designed to mimic the visual appearance of underwater biological fouling using a brown–yellow color scheme, representing different kinds and stages of underwater biological growth. In each iteration of the experiment, an increasing number of patches were added to the net. The entire net was thoroughly filmed after each round of placing the patches. In total, the net was filmed 7 times. Once with no patches, then once with patches covering 22%, 33%, and 44%, and three times with patches covering 66% of the net (but with greater distance to the net each time).

However, a significant change from the previous two years was implemented; instead of manual control during filming, the ROV was autonomously controlled during the underwater missions. This transition from manual to autonomous control allowed for more precise and consistent data collection. The implementation of autonomous control in the controlled ocean-floor pool environment not only facilitated better data collection but also

served to test and validate the developed autonomous control algorithm. The outcomes from this testing phase yielded valuable insights regarding control efficiency, highlighting areas for further improvement or optimization.

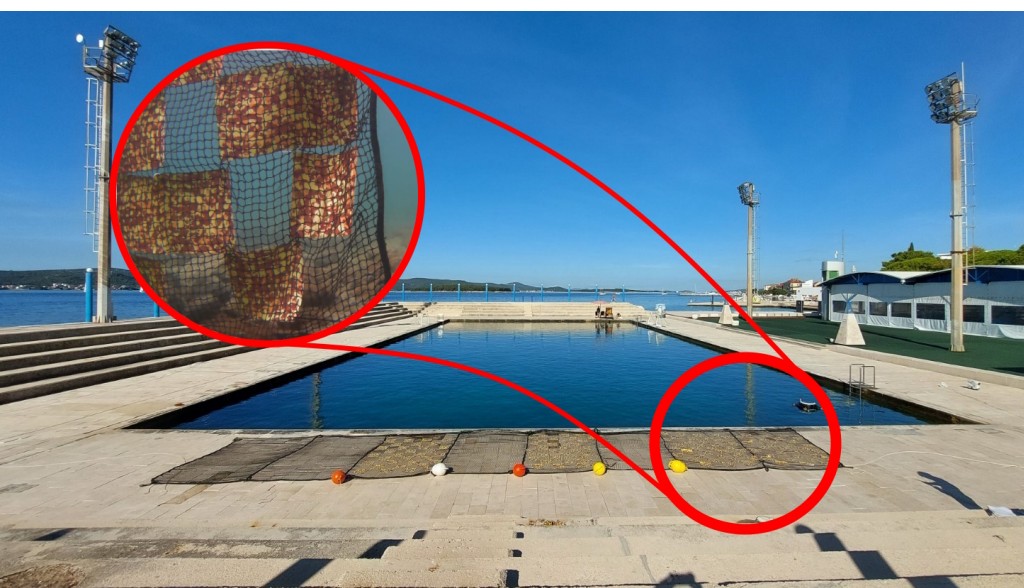

**Figure 10.** The Olympic sea water pool used for testing, with the net dragged out in order to place patches for biofouling simulation. The affixed patches on the net can be seen close-up filmed during a mission in the red circle.

The developed ROV control algorithm was also tested in the pool situated in Biograd. A testing environment like this provided a controlled setting to assess the control algorithm's performance. It also featured an "ocean floor", resulting in more realistic testing conditions as opposed to having a flat artificial floor. The pool's depth varied depending on the tide but generally maintained a depth of approximately 3 m. Using a relatively shallow pool for testing posed a more challenging scenario for the computer vision components of the estimation framework. This is due to the fact that in an actual industrial fish farm the seafloor is not visible due to greater sea depth. The setup of the testing grounds viewed from above is shown in Figure 11.

The deployment of an ASV with the intended SBL acoustic positioning system was not feasible because a pool was used for conducting the experiments. Using a crane to lower and raise the ASV in and out of the pool was not possible. To overcome this limitation, a fixed baseline of transponders was positioned around the pool's edge. These transponders served as reference points for the acoustic positioning system, enabling the accurate localization of the ROV with a transponder within the pool. While this approach differed from the anticipated ASV + ROV combination, it provided a needed practical solution for achieving reliable positioning data during the experimental setup in the pool environment. A screenshot of the UWGPS system GUI can be seen in Figure 12 below, taken during one testing of the ASV+ROV combination. The green line represents the ASV trajectory which should always be available as it has the UWGPS box mounted, while the blue line represents the trajectory of the free-moving transponder that is attached to the ROV in this case, and can travel out of the search range.

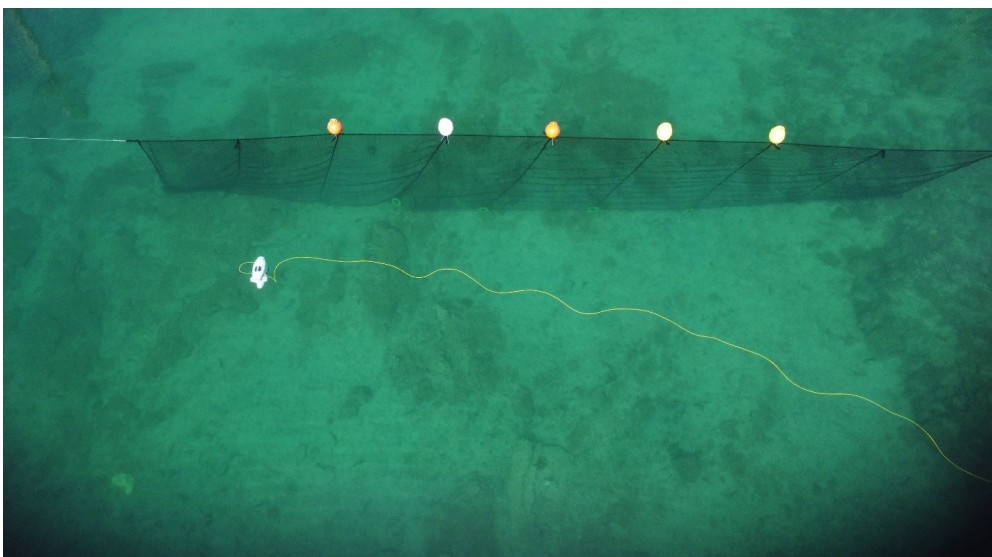

**Figure 11.** Drone shot showing the pool setup experiment in Biograd, Croatia. The net was strung between two posts.

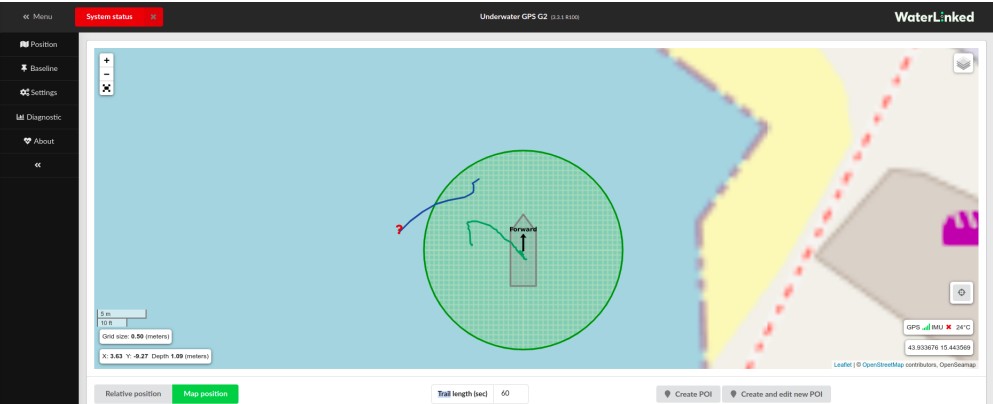

**Figure 12.** Screenshot of the UWGPS GUI while the ROV is tethered to the ASV which also acts as the carrier for the short baseline transponder setup. Since the trajectory of the ROV is visualized, this software was used to roughly estimate the precision of the SBL setup.

## 5. Experimental Results

This section is divided into four subsections, each focusing on different aspects of the study. Section 5.1 provides an in-depth analysis of the autonomous ROV control loop results, presenting the performance, accuracy, and effectiveness of the developed control loop algorithm in enabling autonomous control of the ROV. Section 5.2 delves into the image labeling results, presenting the accuracy and reliability of the labeling process. Section 5.3 discusses the training process and details the UNet model training results. Section 5.4 focuses on the results pertaining to the main research topic, which is the estimation of the amount of biofouling buildup.

### 5.1. Autonomous ROV Control Loop Results

Each ROV mission took around 15 min to cover the aforementioned $42\,\text{m}^2$, and the position is plotted every second to correlate with logged images. The 3D plot in Figure 13 below visualizes the ROV's movement during a mission, and its ideal trajectory generated by the control algorithm. When the 3D plot is flattened to only show the X- and Z-axes like in Figure 14, it is easy to see that the ROV's actual trajectory closely follows the generated one. When only the depth data recorded during a mission are plotted like in Figure 15, it is possible to see that the ROV's automatic depth maintenance system works well in

combination with the developed control algorithm. The depth error tolerance was set to ±0.1 m, and it can be seen that when the error exceeded the set tolerance the ROV was swiftly controlled back into the allowed range. Observing all the autonomous missions using a boxplot to plot the depth error like in Figure 16 one can see that the median value of the depth error is less than 0.05 meters, and almost all of the recorded errors are within the tolerance limit, except a few outlier values, which is expected as the ROV needs a small amount of time to correct the depth.

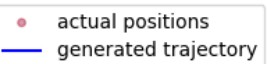

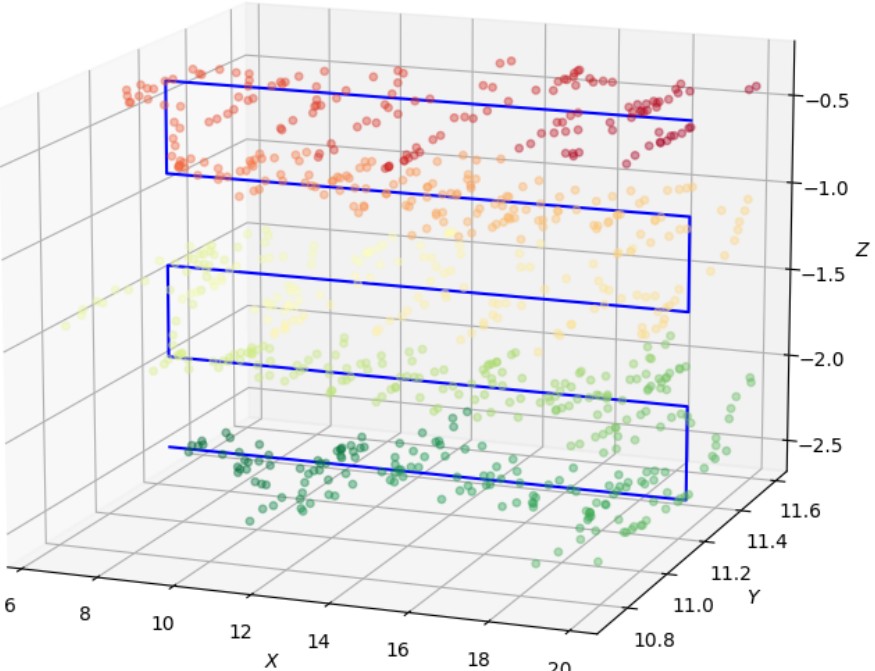

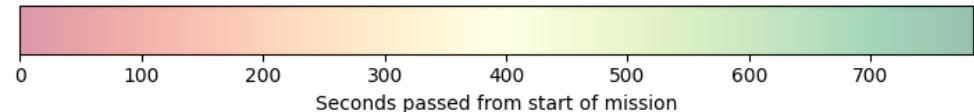

**Figure 13.** 3D plot of the ROV position. The blue line represents a perfect trajectory, the colored dots represent actual positions in time during one mission.

The *Y*-axis in Figure 13 correlates to the distance from the ROV to the net since the net was strung out straight. The controller was implemented with this assumption, so it tried to keep a constant Y-coordinate throughout the mission. This axis was controlled by the surge motion of the ROV. The thrusters dead zone for surge was greater than for heave, which made it more difficult to precisely control the ROV in this direction. The tolerance was doubled to ±0.2 m for this reason. Figure 17 shows the logged data during the course of one mission for the *Y*-axis only. It can be seen that the ROV deviated outside of the tolerance zone more often than for depth, but the controller successfully corrected its behavior each time, bringing it back into the allowed range. The boxplot showing the Y-coordinate keeping error for all the autonomous missions can be seen in Figure 18.

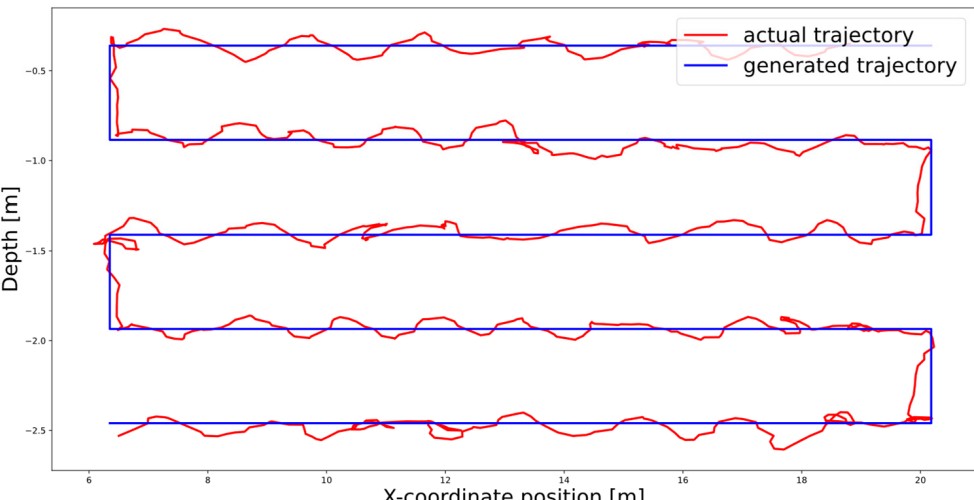

**Figure 14.** Plot showing the recorded trajectory of the ROV compared to the perfectly generated one, during one mission.

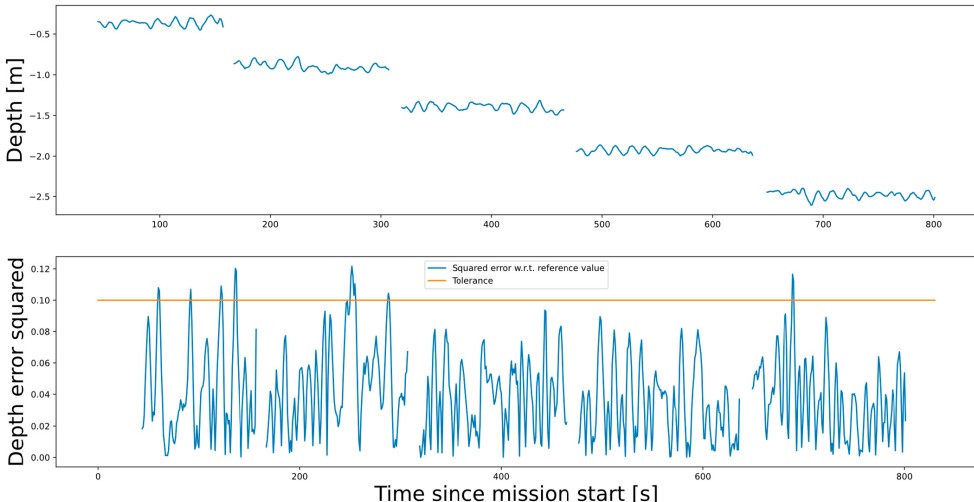

**Figure 15.** Plot showing the error in maintaining wanted depth during purely horizontal movement during one mission. The gaps in the line represent diving movement.

The ROV's SDK has the option to set the gain of the thrusters in order to change the maximum speed. Lowering the gain increases the thrusters sensitivity to low input values, allowing for finer control at the expense of top speed. Seeing how the task of fish pen inspection requires precision of movement and due to the constant filming it would be better to not have sudden changes of direction, the gain was lowered to increase maneuverability. Having the ROV's surge speed be between ∼0.1 m/s and ∼0.2 m/s has shown to be a good compromise between the quality of footage regarding image sharpness, and the duration of a mission. The ROV's velocity data for a single mission can be seen in Figure 19 below.

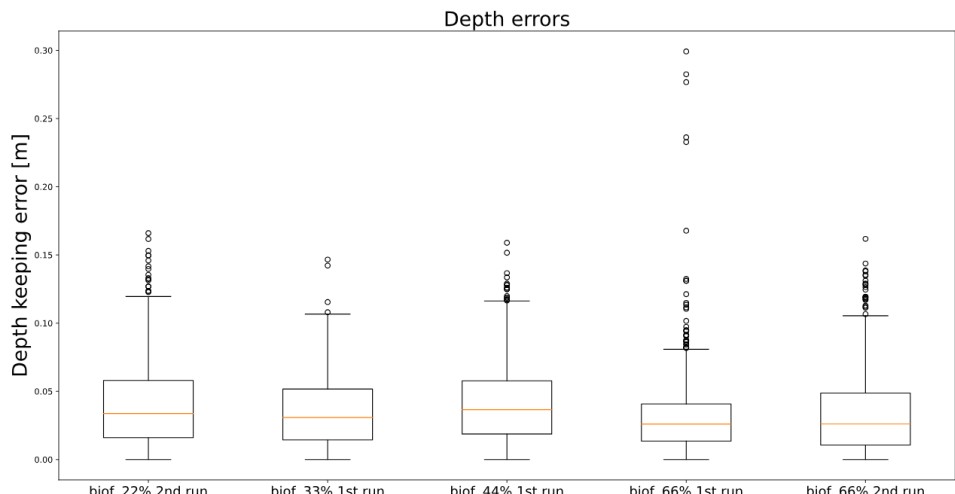

**Figure 16.** Box plot showing the depth keeping error during multiple autonomous missions. The orange line represents the median value of all the values logged. The "box" around the median value captures ±25% of the data. The "whiskers", i.e., the top and bottom line capture the remaining 25% of data. Simply put, each section from line to line represents 25% of recorded data. The dots above the top line represent outliers.

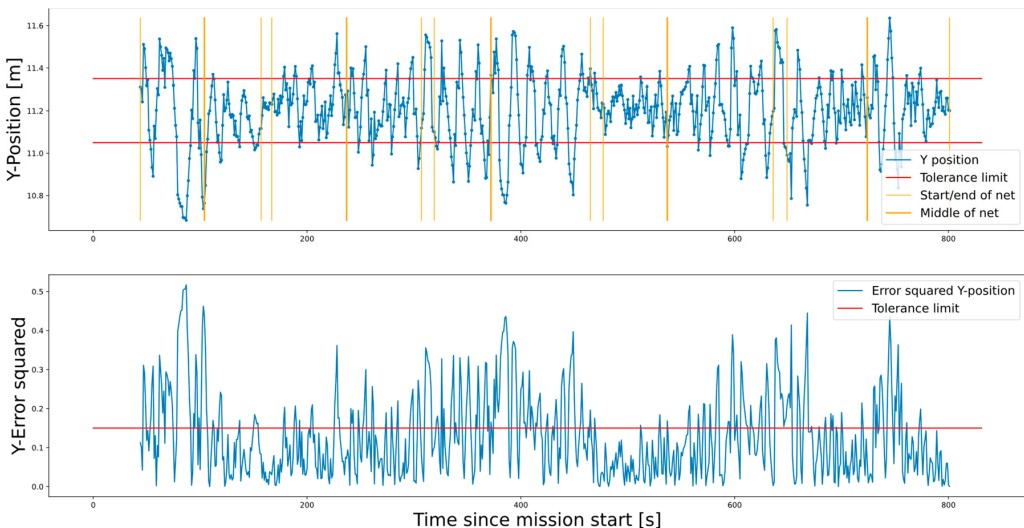

**Figure 17.** Plot showing the Y-position of the ROV throughout one mission.

*5.2. Image Labeling Results*

The images picked for the dataset were represented in the LAB color space. This particular color space was chosen due to its perceptual uniformity, which means that a change in the numerical representation closely follows an actual change in perceived color. The LAB color space separates the luminance (L) channel from the color information (A and B channels). The developed labeling tool was then used to label the images, ensuring precise identification and annotation of the fish pen structures for further analysis and training of machine learning and AI models. The K parameter in K-means clustering algorithm was mostly set to K = 7, but was rarely increased if the labeling precision achieved was not satisfactory. During the image labeling process, only the central part of the full HD image was considered due to concerns related to camera distortion and overall poor image quality near the edges. By focusing on the central portion, which was less affected by distortion and exhibited better image quality, the labeling process could be conducted more reliably, ensuring high-quality annotations and minimizing potential

errors or inconsistencies caused by image imperfections towards the edges. Increasing the K parameter and disregarding image edges helped for footage filmed in Biograd because the sea floor was visible in that footage. This posed an unexpected problem because in the footage filmed at the industrial fisheries at Ugljan the sea floor is not visible due to great sea depth, which is a reasonable assumption to work with when conceptualizing solutions involving computer vision. Moving on, it was necessary to exclude certain labeled images from the dataset after the labeling process was complete. Images exhibiting labeling errors, low image quality, or inconsistencies were identified and removed from the training dataset. This was done to ensure the quality and reliability of the dataset. Table 1 shows the distribution of images in the dataset. A visualization of how images were annotated can be seen in Figure 20.

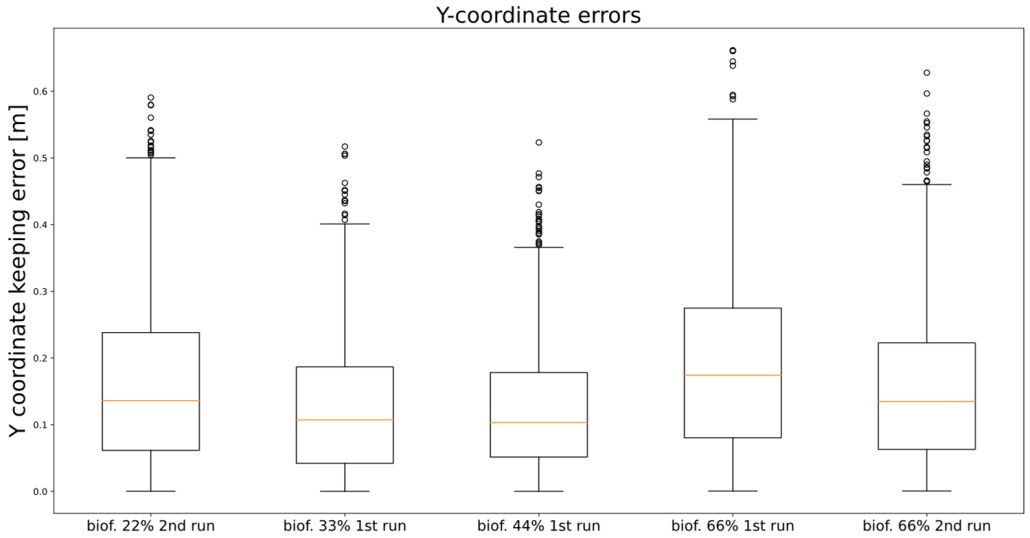

**Figure 18.** Box plot showing the Y-position keeping error during multiple missions. Again as in Figure 16, only a small percent of the recorded data are classified as outliers and shown as dots on the plot.

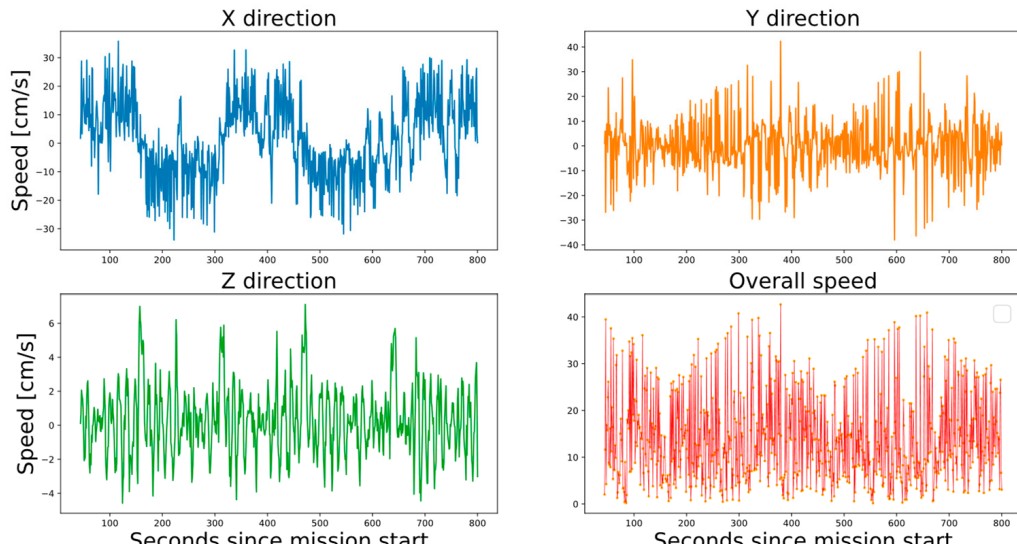

**Figure 19.** Plot showing achieved speeds in the X- (sway), Y- (surge), and Z- (heave) directions during one mission. One can notice the ROV moving left–right and vice versa by examining the X-direction speeds. It is also possible to notice the diving periods of the ROV by the spikes in the Z-direction velocity when reaching the end of the net width.

**Table 1.** Table showing how many images per filming year were labeled in total, and how many of the labeled images ended up in the training/validation dataset for the neural network.

| Footage Year | Location | Labeled Imagess | Images in Dataset |
|:---:|:---:|:---:|:---:|
| 2020 | Ugljan | 938 | 261 |
| 2021 | Ugljan | 405 | 197 |
| 2022 | Biograd | 919 | 694 |

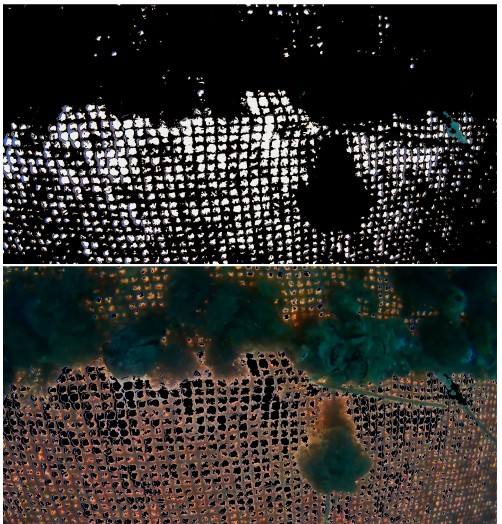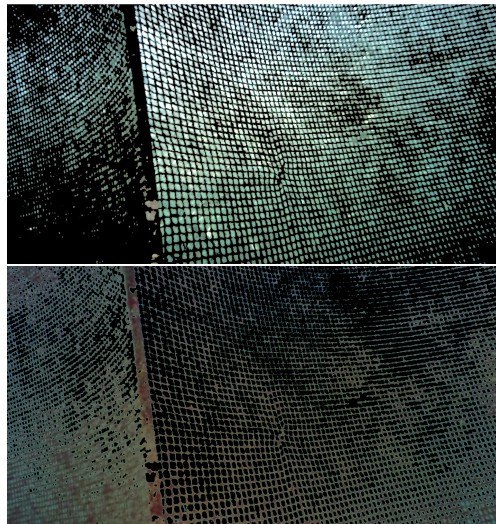

**Figure 20.** Visualization of two labeled images. The top row only has the pixels semantically belonging to the background sea selected, while the bottom row has the pixels belonging to the cage structure selected.

### 5.3. UNet Architecture Segmentation Results

As mentioned earlier, in order to train the UNet model the standard 80–20% train–test split was used, meaning that the model was trained on 80% of available annotated images in the dataset, and the results of the training steps were validated on a randomly selected 20% of images in the dataset that are unseen during training. The learning rate for each training step was automatically adjusted during training as needed, and can be seen in Figure 21 below. The training process was stopped once no more discernible improvement was shown from one iteration to the next.

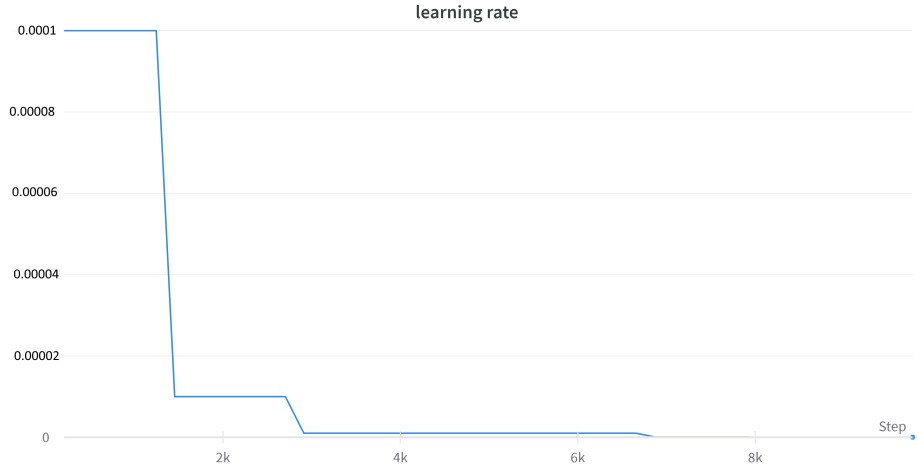

**Figure 21.** Plot showing the learning rate used in each step of the training.

Dice score is a common metric used for scoring the performance of image segmentation models, ranging from 0.0 to 1.0 (a higher value is better). It measures the similarity or overlap between the predicted segmentation mask and the ground truth segmentation mask [23]. The highest Dice score achieved on the validation set of images was 0.8434 during the 9th training epoch, and the weights calculated to achieve this coefficient were saved to be used later on. The Dice score can be seen changing during the training process in Figure 22. A visualization of how the trained UNet model successfully segments the images can be seen in Figure 23.

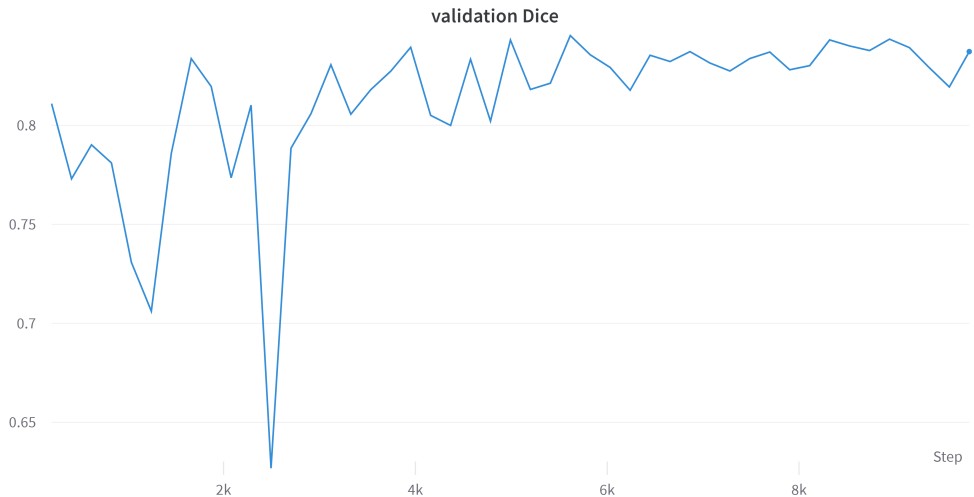

**Figure 22.** Plot showing the popular Dice score used in image segmentation analysis and its change during the training.

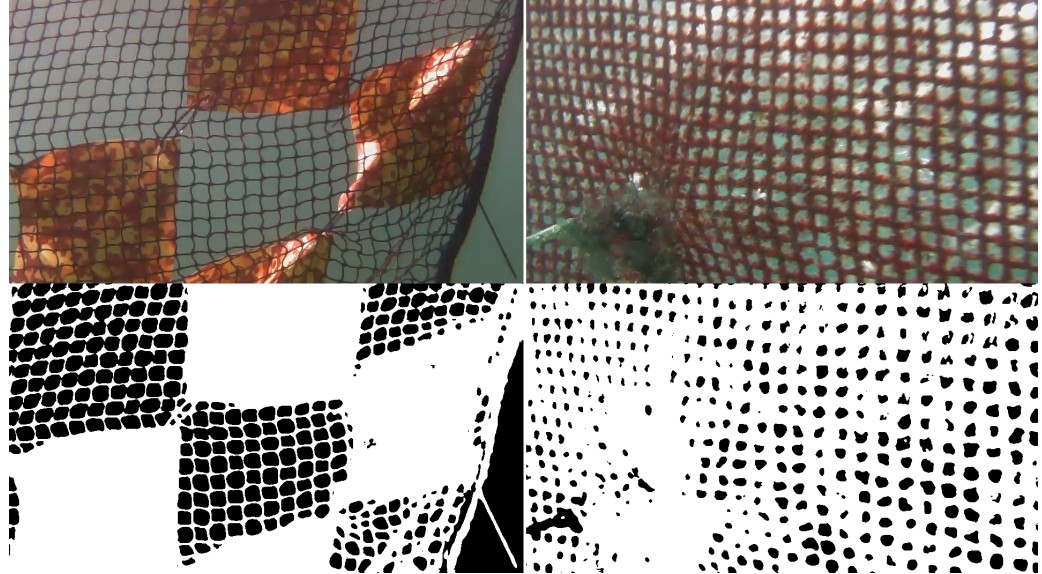

**Figure 23.** Result of predictions made by a trained UNet neural network segmentation model for: (**left**)—an image taken in the controlled conditions in Biograd, (**right**)—an image taken at a real fishery near Ugljan.

### 5.4. Biofouling Estimation Results

As previously mentioned, an industrial fishery provided a fish pen net for the experimental setup. Square patches were affixed onto the net in a series of iterations to simulate biofouling. The patches were incrementally added in four stages, progressively covering larger areas of the net. All of missions had the ROV at around 1 m away from the net being

filmed, except a repeated mission at 66% where the distance was purposefully increased to around 1.5 m.

To establish a benchmark result, the biofouling estimation algorithm was initially applied without the implementation of any filtering methods mentioned in Section 3.2.2. All of the recorded footage in a filming session was only cropped around the center of the frame and fed into the estimation algorithm. Table 2 shows the benchmark estimated percentages.

**Table 2.** Table showing the actual simulated biofouling percentage, and the estimated biofouling percentage generated by the estimation framework.

| Actual Biofouling | Estimated Biofouling |
|---|---|
| 22.00% | 16.00% |
| 33.00% | 32.19% |
| 44.00% | 41.02% |
| 66.00% | 65.48% |

5.4.1. Framework Filters Results Using UNet for Semantic Segmentation

The two filter methods implemented were the exclusion of footage during non-uniform movement, and the contour filtering based on size and shape of detected contours. As can be seen in Figure 24 below, the filters made only a small difference to the estimated percentage. This is due to the fact that the autonomous filming worked well in a sense that the ROV's speed was consistent throughout the mission and the angle of the filming was good. Each area of the net is filmed for around the same amount of time, and the neural network semantic segmentation model performed well, so the benchmark result without filtering was close to correct from the start.

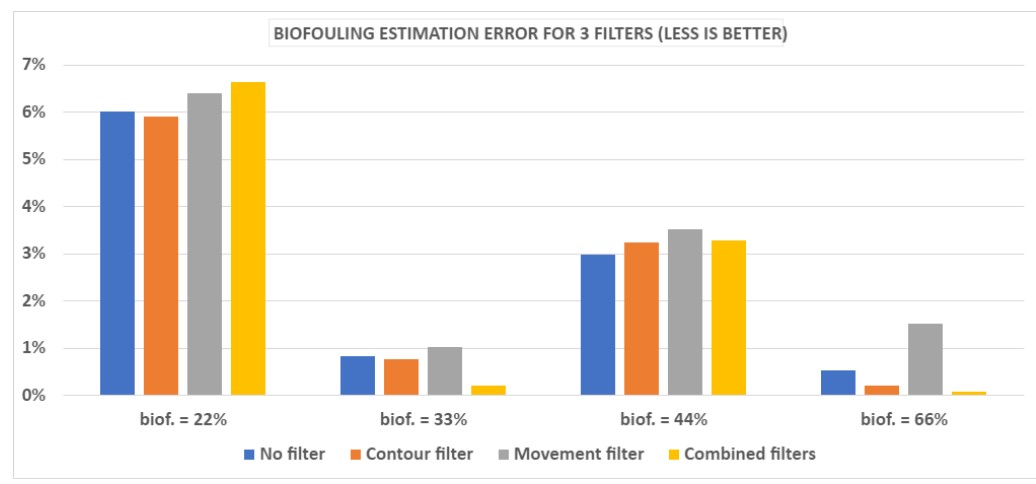

**Figure 24.** Bar plot showing the error in the estimated percentage of biofouling.

When the filming conditions are not perfect, such as in the repeated 66% coverage scenario tested in Biograd, then the filters help out. The filming conditions were purposefully worsened by having a greater distance from the ROV to the net during filming. The combination of both filters reduced the estimation error by 1.75% in total, as can be seen in Table 3 below.

**Table 3.** Table showing the actual simulated biofouling percentage, and the estimated biofouling percentage generated by the estimation framework with the combinations of the footage and computer vision filters turned on.

|  | Value | Biofouling Estim. Error |
|---|---|---|
| Actual biofouling | 66.00% | / |
| No filter estim. | 48.04% | 17.96% |
| Contour filter estim. | 49.44% | 16.56% |
| Movement filter estim. | 48.98% | 17.02% |
| Combined filter estim. | 49.79% | 16.21% |

5.4.2. Effect of Filming Conditions

As can be seen in Table 3, when the filming conditions are unfavourable the estimation error is unacceptably large. This was first noticed during a control run while testing the autonomous movement. The net had no patches affixed to it meaning that the estimated percentage of biofouling should be close to 0%. However, the heading angle of the ROV was fixed to a constant value. The filming angle during the mission was not always perfectly perpendicular in relation to the net, since the net was strung between two posts and the bottom of the net could not be fixed to anything. This made the net slightly convex and wavy, as opposed to the tightly strung out fish pen nets at fisheries that are almost perfectly straight, except at the rounded bottom. Also, there is the possibility for the heading to slowly drift on its own during a mission because of an imperfectly calibrated compass built into the ROV. The heading drift coupled with the slightly wavy net produced an estimation of 32.37% biofouled net area, when it should have been 0%. The clean net was filmed from an otherwise good distance of ∼1 m, but with a tilted angle.

One of the considerable factors during filming is the time of day, that is, the position of the Sun. The used Blueye Pro ROV does not have an HDR camera, meaning that the overexposure of one part of the camera sensor to light ruins the rest of the image. This effect of course decreases as depth increases, but still poses a problem when filming near the surface. Filming missions should be planned accordingly, holding them early in the morning or late afternoon, or filming the cages with the Sun behind the camera. Furthermore, it goes without saying that the filming distance greatly impacts image quality and the estimation process as a whole. Filming should be done at a distance where the the net can be clearly separated from the background, i.e., the edges should be sharp and easily discernible.

As mentioned earlier in Section 5.2, the sea floor could have a big impact on the computer vision component of the framework. Having the sea floor visible increases the already difficult challenge of accurate semantic image segmentation. The scenario is unlikely because fish farms should be situated 3km away from shore and have 50 m of depth available to limit the environmental impact, as mentioned in [24,25].

5.4.3. Choosing the UNet Architecture Instead of Logistical Regression

It is important to note that the results seen so far were achieved using the trained UNet architecture model for semantic image segmentation. While logistical regression initially seemed promising for semantic segmentation (previous research done in [10]), its limitations became evident during testing in Biograd. It quickly became apparent that the method is not robust enough. Small changes in lighting conditions completely threw off the segmentation which then produced unusable results. Adding the footage captured in Biograd into the training dataset does improve estimation results, but at the same time it degrades the quality of segmentation from footage captured in previous years. Still, the biofouling estimation algorithm was run with both the old and new (added footage from Biograd) logistical regression models, and the results can be seen in Table 4 below. All of the estimated percentages in the table are generated by the framework without using any of the

filters mentioned before. In Figure 25, below that shows the absolute estimation error for each autonomous filming mission, it is apparent that using logistical regression not trained on new footage produces much more inaccurate results than the other two models. Training a logistical regression model with new footage does improve performance, but at the cost of overtraining which can be seen by poor performance for the highest biofouling scenario. The results for that particular test are worse because the filming conditions are different in a sense that the previous three filming missions were carriedo ut on Wednesday afternoon, and the 66% biofouling coverage filming was done the next day on Thursday morning.

**Table 4.** Table showing the average estimate of biofouling percentage when using the footage filters, for each semantic segmentation model.

| Actual Biofouling | Log. Reg. Estim. (Old Footage) | Log. Reg. Estim. (New Footage) | UNet |
|---|---|---|---|
| 22% | 29.96% | 20.99% | 16.00% |
| 33% | 50.12% | 34.63% | 32.19% |
| 44% | 53.49% | 41.73% | 41.02% |
| 66% | 52.45% | 50.05% | 65.48% |

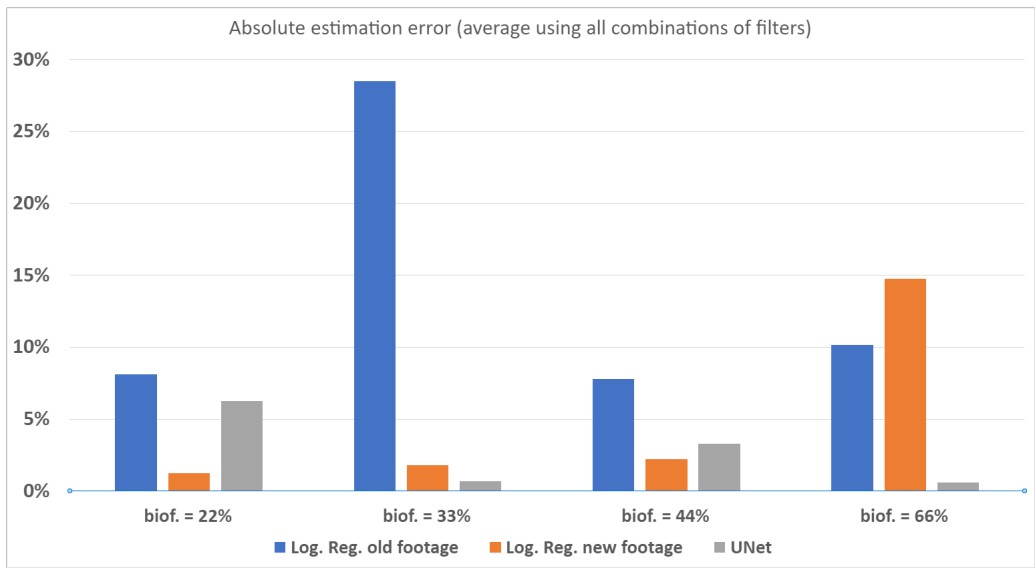

**Figure 25.** Bar plot showing the average absolute error in the estimated percentage of biofouling for logistical regression models trained on just old (blue bar) and combined (orange bar) data, and UNet.

To sum up, using a robust UNet model for the image segmentation node in the biofouling framework produced an average absolute biofouling estimation error of just 2.54%, as can be seen in Figure 25. The image segmentation is the most computationally heavy task of the framework, and the time taken to segment an image using a trained UNet model depends on the power of the onboard computer in the ASV if the estimation is to be done in real-time, or the workstation if the estimation is to be carried out after filming is complete. Using a dedicated GPU to run the UNet model is heavily recommended because it reduces the processing time of a 960 × 540 resolution image from a few seconds using a CPU, to the millisecond order of magnitude using a GPU. Furthermore, using UNet for segmentation means that potentially obtaining a diverse range of footage from various fisheries filmed under different conditions could only enhance the model's robustness and performance, whereas opting for simpler models like logistical regression would lead to overtraining when faced with varying scenarios.

### 6. Conclusions and Future Work

In conclusion, the main contributions of this research were: (1) development of a labeling tool in order to create a curated dataset of labeled underwater HD images of fish pens, (2) development and implementation of a framework successfully used for estimating the amount of biofouling on the nets of fish pens, that incorporates a trained AI neural network model used for the task of semantic segmentation of underwater images of fish pens, and (3) development of an autonomous closed-loop control system using the available SDK for the ROV and the available localization data supplied by the retrofitted underwater positioning system.

The control loop algorithm successfully controlled the ROV in a pool setting using point-to-point navigation. Although the scenario in which the experiment was conducted is not an exact replica of the actual conditions in a fish farm, it still demonstrates the possibility of using the ROV as an autonomous vehicle to perform inspection missions. The control algorithm could be modified to include a three-dimensional map of the farm and also fuse SONAR measurements if such a sensor would be mounted onto the ASV, together with UWGPS and live camera footage, to achieve localization in a complex environment. The labeling tool made it possible to accurately segment and semantically label the images of fish pens at around 30 s or less per image, thus allowing us to create a dataset of more than a thousand images within a satisfactory time frame. Perhaps most important, the implementation of the proposed framework which was tested in a controlled environment proved to be a success with the absolute value of the estimation error roughly being 2.5%. It is also worth noting that carrying out the inspection mission in one take while keeping the velocity of the ROV almost constant throughout, without much backtracking or spending too much time filming one area in relation to the rest of the net, produces an accurate estimation. Precisely determining filming positions to keep the overlap of footage fed to the estimation framework at a minimum might not be cost-effective to develop. Due to the nature of the current inspection process which involves divers estimating the state of the net, a "good enough" estimation is satisfactory.

For future endeavors, the developed framework should be tested at an industrial fishery using the original idea of pairing the ASV Korkyra with an ROV. This field testing would aim to validate the framework's effectiveness in a challenging environment like an industrial fish farming operation. Furthermore, the previously developed proprietary tether management system (cited in [14]) that also relies on localization of the ROV should be integrated physically onto the ASV, and its driver software should be fully tested in such a setup. Lastly, the project also envisions an air surveillance aspect using a light autonomous drone. The landing platform for the drone was designed in such a way that it can be mounted onto the ASV Korkyra [13]. The combined heterogeneous system comprising the ASV Korkya, an ROV operating with the TMS, and the drone should be tested in a real-world scenario in the future.

**Author Contributions:** Conceptualization, N.K. and N.M.; methodology, N.K. and M.F.; software, M.F.; validation, N.K. and M.F.; investigation, N.K. and M.F.; resources, N.K. and M.F.; data curation, N.K. and M.F.; writing—original draft preparation, M.F.; writing—review and editing, N.K. and M.F.; visualization, N.K. and M.F.; supervision, N.K. and N.M.; project administration, N.M. and N.K.; funding acquisition, N.M. All authors have read and agreed to the published version of the manuscript.

**Funding:** Research work presented in this article has been supported by the project Heterogeneous autonomous robotic system in viticulture and mariculture (HEKTOR) financed by the European Union through the European Regional Development Fund-The Competitiveness and Cohesion Operational Programme (KK.01.1.1.04.0036); the ERDF-funded project (KK.01.1.1.07.0069) Multifunkcionalne pametne bove (Multifunctional smart buoys); the "Razvoj autonomnog besposadnog višenamjenskog broda" project (KK.01.2.1.02.0342) co-financed by the European Union from the European Regional Development Fund within the Operational Program "Competitiveness and Cohesion 2014–2020". The content of the publication is the sole responsibility of the project partner UNIZG-FER; and by

ONR Robot Aided Diver Navigation in Mapped Environments—ROADMAP project under Grant Agreement No. N000142112274.

**Institutional Review Board Statement:** Not applicable.

**Informed Consent Statement:** Not applicable.

**Data Availability Statement:** The data that support the findings of this study are available from the corresponding author, M.F., upon reasonable request.

**Conflicts of Interest:** The authors declare no conflict of interest.

## Abbreviations

The following abbreviations are used in this manuscript:

| | |
|---|---|
| AI | Artificial intelligence |
| ASV | Autonomous Surface Vehicle |
| AUV | Autonomous Underwater Vehicle |
| CPU | Central Processing Unit |
| FOV | Field of View |
| GPS | Global Positioning System |
| GPU | Graphical Processing Unit |
| GUI | Graphical User Interface |
| HD | High Definition |
| HEKTOR | Heterogeneous Autonomous Robotic System In Viticulture And Mariculture |
| LIDAR | Light Detection and Ranging |
| PID | Proportional–integral–derivative |
| RF | Radio Frequency |
| ROS | Robotic Operating System |
| ROV | Remotely Operated Underwater Vehicle |
| SBL | Short Baseline |
| SDK | Software Development Kit |
| SONAR | Sound Navigation and Ranging |
| UAV | Unmanned Aerial Vehicle |
| UML | Unified Modeling Language |
| UWGPS | Underwater GPS |

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
