# Peer review of "Autonomous Visual Fish Pen Inspections for Estimating the State of Biofouling Buildup Using ROV"

_jmse, doi:10.3390/jmse11101873_

Round 1
Reviewer 1 Report
GENERAL 1. What is the main question addressed by the research? Algorithms for ROV & image processing for fish pen. 2. Do you consider the topic original or relevant in the field? Does it address a specific gap in the field? Application is original, but the algorithms are not. 3. What does it add to the subject area compared with other published material? Application of fish pen. 4. What specific improvements should the authors consider regarding the methodology? What further controls should be considered? See comments below. 5. Are the conclusions consistent with the evidence and arguments presented and do they address the main question posed? Yes. 6. Are the references appropriate? See comments below. MORE COMMENTS 1. Title - it is suggested to delete "an" 2. Abstract - too many keywords. It is suggested to have 5 keywords 3. Abstract - Need to include problem statement just before objective 4. Abstract - Need to include a brief conclusion after results description 5. Introduction - this section is too long. It is suggested to summarize some contents such as HEKTOR, etc. 6. Introduction - literature review must be written before research gaps, significant of research/contribution & objective. 7. Methods - Control algorithm & image processing algorithm are not clearly defined. Please include more details regarding the algorithm. 8. Methods - K-means clustering & UNet are not clearly defined. Please include more details regarding the algorithm. 9. Methods - What is the % arrangement of training, testing & validation for AI? 10. Methods - Pre-processing methods are not clear. What methods have been used? 11. Results - Method, set up must be written in Methods section. Not in Results section. 12. Results - There is no result for segmentation accuracy & pre-processing (MSE, PSNR, SSIM, etc). Please include them. 13. Conclusion - Figure & Table in this section must be moved to results section. 14. Reference - 7/20 = 35% out-of-date. However, this is acceptable.Author Response
Please see the attachment.

Reviewer 2 Report
I read the paper several times But I could not find any writings related to the following Figures: 5,8,9,10,11 and 12. Please see attachment.

Round 2
Reviewer 1 Report
All comments have been addressed.